# Safety and Efficacy of HU-014 in the Treatment of Post-Stroke Upper Limb Spasticity: A Phase I Pilot Study

**DOI:** 10.3390/toxins14110730

**Published:** 2022-10-25

**Authors:** Junekyung Lee, Min Ho Chun

**Affiliations:** Department of Rehabilitation Medicine, Asan Medical Center, University of Ulsan College of Medicine, Seoul 05505, Korea

**Keywords:** botulinum toxin, safety, stroke, muscle spasticity, upper extremity

## Abstract

Botulinum toxin type A (BTX-A) is widely used for treating post-stroke upper limb spasticity. We evaluated the safety and efficacy of HU-014 in treating post-stroke upper limb spasticity. Thirteen patients were administered with HU-014. The primary outcome was safety, including adverse events, vital signs, physical examination, laboratory tests, and antibody formation test. The secondary outcomes were changes in the Modified Ashworth Scale (MAS) score for wrist, elbow, and finger flexor; Disability Assessment Scale (DAS); Investigator’s Global Assessment (IGA) and Subject’s Global Assessment (SGA); Caregiver Burden Scale (CBS); and Columbia Suicide Severity Rating Scale (C-SSRS) at weeks 4, 8, and 12 from baseline. No notable safety-related issues were reported. MAS and DAS scores were significantly decreased from those at baseline at 4, 8, and 12 weeks (*p* < 0.05). At weeks 4, 8, and 12, the IGA and SGA scores were 5.85 ± 0.55, 5.69 ± 0.48, and 5.62 ± 0.65 and 5.46 ± 1.20, 5.85 ± 0.38, and 5.77 ± 0.73, respectively. CBS scores decreased at all timepoints and those for cutting fingernails significantly decreased at 8 and 12 weeks compared with baseline (*p* < 0.05). C-SSRS scores showed that suicidal ideation in all patients was “low” at all timepoints. HU-014 is a safe treatment that can improve post-stroke upper limb spasticity.

## 1. Introduction

Spasticity occurs in 4–42.6% of patients after stroke and can cause disability, pain, and secondary complications such as impaired movement and joint contracture, thereby reducing quality of life and increasing the burden of caregivers [1,2]. Botulinum toxin type A (BTX-A) is commonly administered to patients with post-stroke upper limb spasticity [3] and has been proven to be safe and effective [3,4].

Effective management is important for patients with upper limb spasticity after stroke because spasticity and abnormal postures cause discomfort for patients, difficulties for caregivers, and ultimately reduced quality of life [5,6]. Treatment methods for spasticity include physical therapy, electrical stimulation, oral antispastic medications, neuromuscular blockade with phenol or alcohol and botulinum toxin, and surgical treatment [6]. While oral antispastic drugs are considered for generalized spasticity, side effects such as sedation, drowsiness, dizziness, weakness, and confusion frequently occur. Patients with focal spasticity are usually injected with intramuscular botulinum toxin [6], which inhibits the secretion of acetylcholine from the presynaptic endplate acting on the neuromuscular junction, thereby lowering muscle tone and relieving spasticity after stroke [7].

HU-014 (Liztox^®^, Huons Biopharma, Seongnam, Korea) is a newly developed BTX-A. Botulinum toxin dose is measured in units (U), based on the median lethal dose (LD50) of the neurotoxin [8]. One unit of botulinum toxin is the dose of intraperitoneally injected toxin found to kill 50% (LD50) of a group of mice. The biological potency of HU-014 was based on the LD50 after the mouse LD50 assay. In the in vivo test, the response to electrical stimulation is measured after HU-014 is administered to the hindlimb muscles of mice, resulting in a decrease in the electrical stimulation response in a dose-dependent manner; the response was similar to that of onabotulinum toxin A (Botox^®^, Allergan Inc., Irvine, CA, USA). HU-014 has toxicity symptoms similar to those of BTX-A developed and commercially available; it also has no specific toxicity.

HU-014 is currently marketed for the improvement of glabellar lines. However, its safety and efficacy have not been confirmed in patients with post-stroke upper limb spasticity. In this study, we aimed to investigate the safety and efficacy of HU-014 in the treatment of post-stroke upper limb spasticity.

## 2. Results

### 2.1. Demographic and Baseline Characteristics

Figure 1 shows the study flowchart. Thirteen patients underwent screening tests. All 13 patients registered in this study received HU-014, and two (15.38%) of them dropped out of the trial due to voluntary withdrawal of consent. Finally, 11 patients (84.62%) completed this study.

Demographic and baseline characteristics of participants are presented in Table 1.

### 2.2. Safety Measures

Injection doses per muscle are presented in Table 2. There were no non-TEAEs during the study period. TEAEs occurred in three patients (23.08%, 7 cases) out of 13 patients in the safety set analysis: the most frequent was “vertigo positional” in one patient (7.69%, 2 cases), followed by “diabetes mellitus,” “hyperglycemia,” “iron deficiency anemia,” “periodontal disease,” “insomnia” in one patient each (7.69%, 1 case). All of these were systemic adverse events not associated with the injection site of HU-014 (Table 3). The severity of the seven TEAEs was mild in three TEAEs, including “vertigo positional,” “hyperglycemia,” and “periodontal disease,” and moderate in four TEAEs, including “iron deficiency anemia,” “vertigo positional,” “diabetes mellitus,” and “insomnia.” In the causal relationship with HU-014, six TEAEs (except “iron deficiency anemia”) were confirmed as “definitely not related, none.” Of the 13 patients, only 1 (7.69%, 1 case) presented with ADR, among TEAEs, which was moderate “iron deficiency anemia.” A therapeutic drug was administered for the treatment of the ADR, and it was later confirmed as “recovered/resolved.” In addition, the causal relationship with HU-014 was judged as “unlikely, probably not related,” which might be due to the patient’s history of iron deficiency anemia. Of the 13 patients, only one (7.69%, 1 case) presented with SAE, moderate “vertigo positional.” The causal relationship with HU-014 was confirmed as “definitely not related, none,” and as “recovered/resolved” as it recovered without special treatment. No adverse events or ADR leading to death or discontinuation of HU-014 were noted during the study period.

No clinically significant abnormalities were observed in the vital signs (blood pressure, heart rate, body temperature). None of the patients showed a clinically significant change in abnormal findings after administration of the HU-014 compared to the baseline. Results of the antibody formation test confirmed that no patients were “positive” at all time points.

In the laboratory tests, one patient showed an abnormal clinically significant change after administration of HU-014 compared to baseline, which was reported as an adverse reaction (hyperglycemia). Subsequently, the patient was diagnosed with “diabetes mellitus,” which was additionally reported as an adverse event, and follow-up for “hyperglycemia” was terminated. For both adverse reactions, the causal relationship with HU-014 was evaluated as “definitely not related, none.” Treatment drugs (metformin, vildagliptin, and glimepiride) were administered to treat diabetes mellitus, and follow-up was terminated because further follow-up was deemed unnecessary.

### 2.3. Efficacy Measures

For efficacy analysis, FAS was the main analysis group and PPS was the auxiliary group. The final judgment on the efficacy endpoint was made through the FAS.

At weeks 4, 8, and 12 after HU-014 administration compared to the baseline, the changes in the MAS of each muscle were as follows: the wrist flexors, −1.31 ± 0.48, −1.19 ± 0.56, and −1.08 ± 0.49 (*p* = 0.0002, 0.0005, and 0.0005); the elbow flexor, −0.50 ± 0.50, −0.58 ± 0.49, −0.46 ± 0.59 (*p* = 0.0078, 0.0012, and 0.0160); and the finger flexors, −1.50 ± 0.61, −1.38 ± 0.74, and −1.42 ± 0.67 (*p* = 0.0005, 0.0010, and 0.0002). All results demonstrated significant reduction at all time points (Figure 2).

Changes in the target items in the DAS scores at 4, 8, and 12 weeks after the intervention from baseline were −1.23 ± 0.60, −1.15 ± 0.69, and −1.15 ± 0.69, which also decreased significantly at all time points (*p* = 0.0005, 0.0010, and 0.0010; Table 4).

At 4, 8, and 12 weeks after intervention, the IGA score was 5.85 ± 0.55, 5.69 ± 0.48, and 5.62 ± 0.65, respectively, and the SGA score was 5.46 ± 1.20, 5.85 ± 0.38, and 5.77 ± 0.73, respectively (Figure 3).

At 4, 8, and 12 weeks after intervention compared with the baseline, the change in CBS scores for cleaning the palm, dressing, and cleaning under the armpit decreased, but not statistically significant (*p* > 0.05) as well as for cutting the fingernails, with a significant difference at 8 and 12 weeks (*p* < 0.05), but none at 4 weeks (Table 5).

The suicidal ideation in C-SSRS of all patients was “low” at baseline and 4, 8, and 12 weeks after the intervention (Table 6). No patients had any changes in their suicidal ideation of C-SSRS at any time point post-intervention compared to that at baseline.

## 3. Discussion

This study is the first phase I clinical trial evaluating the safety and efficacy of HU-014 in the treatment of post-stroke upper limb spasticity. After HU-014 administration, there were no noticeable problems with safety, but muscle spasticity, degree of disability, and caregiver burden in patients with post-stroke upper limb spasticity were reduced; both the investigator and the patients showed overall satisfaction with the improvement of symptoms during the study period. Therefore, HU-014 exhibits the effect of improving upper limb spasticity after stroke.

BTX-A is effective and safe for the treatment of post-stroke upper limb spasticity [3,4,9]. Regarding safety, which was the primary outcome of this study, no safety-related issues were noted and no significant abnormalities were observed in vital signs or physical examination. All adverse events reported in this study were mild or moderate, and no adverse events leading to death occurred. In the antibody formation test, no patients developed antibodies to BTX-A at the administered dose of HU-014 (up to 360 U) during the study period. No specific findings were noted compared with the safety of BTX-A reported in previous studies in the treatment of post-stroke upper limb spasticity [10,11,12,13,14,15].

The recommended injection doses and sites used in this study followed those of previous studies on the efficacy and safety of botulinum toxin type A in the treatment of post-stroke upper limb spasticity [10,11,16,17]. Consistent with the methods adopted in previous studies, the dose for injection and target muscles were selected by the physician based on the degree of spasticity. The actual injection dose of HU-014 used in this study is shown in Table 3. The injection dose varies depending on the patient’s spasticity; however, this study is not a dose-range study (i.e., it is not a clinical trial in which different doses of an agent are tested to determine which dose is most effective and/or least harmful).

A wide range of dilution ratios may be used for post-stroke upper limb spasticity [18]. A dilution of 25 units/mL of HU-014 was used in this study. For onabotulinum toxin A, a dilution of 50 units/mL was most commonly used, but higher dilutions of 25 units/mL and 20 units/mL were also used in several studies [7,19]. Various dilutions such as 20, 50, and 100 units/mL were also used for incobotulinum toxin A [20]. It is assumed that a higher dilution of BTX-A may result in better therapeutic effects than a lower dilution volume; however, this point remains controversial [7,19,20]. Therefore, the implications for using higher dilutions of 25 units/mL of HU-014 than those used in pivotal studies with onabotulinum toxin A or incobotulinum toxin A remain unclear. Further research is needed to identify the implications of using a higher dilution of BTX-A.

At all time points after HU-014 administration compared to the baseline, muscle spasticity of the wrist flexor, elbow flexor, and finger flexor measured by MAS were significantly reduced, showing an improvement in muscle spasticity. The effect persisted until 12 weeks after HU-014 administration, which is consistent with the results of previous studies [16,21,22,23]. Since a decrease in MAS score of 1 point or more from the baseline indicates clinical significance [12,24], then, in this study, the muscle spasticity of the wrist flexors and finger flexors was significantly improved. The relatively small decrease in muscle spasticity of the elbow flexors might be because muscle spasticity in the elbow flexors at baseline was lower than in the wrist flexors and finger flexors, and the dose of the drug injected into the elbow flexor was relatively small based on the maximum recommended dose.

In the present study, the DAS score showed a significant improvement at all time points for 12 weeks, which is also consistent with the results of previous studies that showed upper limb function measured by DAS was improved after BTX-A administration [11,16,17]. Global assessment evaluated by the investigator and patients was an average of 5 points or more at all time points, which suggested that most patients were satisfied with the treatment. Compared to the baseline, the CBS score of each item at all time points showed a tendency to decrease after HU-014 administration. This was also consistent with previous findings [11,17].

Spasticity is a motor disorder characterized by a velocity-dependent increase in muscle tone or a tonic stretch reflex associated with hypertonia [25]. The evaluation of spasticity in this study includes an assessment of the velocity-dependent increase in resistance to passive movements. We measured spasticity using the MAS, which is the most commonly used tool for quantifying spasticity [26]. Regarding disabling spasticity, there is no generally accepted definition of this condition [2]. Disabling spasticity was defined as spasticity that requires interventions such as physiotherapy and orthoses or pharmacological treatment. It was also defined by Lundström as spasticity that affects movement function, activity performance, or participation in social life, accompanied by positive symptoms of upper motor neuron syndrome [2,26]. Therefore, disabling spasticity requires a comprehensive clinical evaluation of patients with regard to the presence and impact of spasticity from a disability perspective. The MAS used to measure spasticity in this study has the limitation that it measures only increased muscle tone. Disability from spasticity was not directly evaluated in this study, but changes in passive function were assessed by including DAS as a secondary outcome measure.

## 4. Study Limitations

This study has some limitations. First, the sample size was relatively small. As this was a phase I pilot study, it was conducted with the minimum number of subjects empirically required. A future study with a large sample size is needed to sufficiently demonstrate the effectiveness of HU-014. Second, the study period was relatively short. Future studies with a larger sample size and long-term follow-up are required to evaluate the long-term safety and efficacy of the drug. Third, this study did not have a control group. Although BTX-A has been demonstrated to be more effective than a placebo for treating post-stroke upper limb spasticity [3], further studies need to directly compare HU-014 with controls. We plan to conduct phase II or III studies on the safety and efficacy of HU-014 with a larger sample size and long-term follow-up and perform a comparison with controls. Finally, in this study, only reported adverse effects were considered, and this study did not search for additional adverse effects with a checklist, such as the TOWER study, a dose escalation study with inco-BoNT, did. As well this study did not include measurements of pulmonary function, such as the TOWER study, such as forced expiratory volume in 1 s and maximal inspiratory pressure [27].

## 5. Conclusions

HU-014 showed no safety issues and was effective in treating post-stroke upper limb spasticity. Therefore, HU-014 is safe for improving upper limb spasticity in patients with stroke.

## 6. Materials and Methods

### 6.1. Study Design

This prospective, single-center, phase I pilot study was performed between December 2019 and May 2020 at Asan Medical Center in the Republic of Korea, registered at clinicaltrials.gov (NCT04415346), and approved by the Institutional Review Boards of our hospital. All patients gave written informed consent prior to study enrollment. As this is a sponsored study, the sponsor developed the study protocol of this trial, but the researchers conducted the study process, manuscript formation, and submission.

### 6.2. Participants

The inclusion criteria were as follows: (1) age ≥ 19 years, (2) diagnosis of stroke at least 6 weeks prior to screening, (3) modified Ashworth Scale (MAS) score with the wrist flexor score ≥2 and at least one of the elbow flexor and finger flexor score ≥1, and (4) Disability Assessment Scale (DAS) score ≥ 2 in one of the categories of hand hygiene, dressing, upper limb position, or pain for evaluation. The exclusion criteria were as follows: (1) history of hypersensitivity to ingredients included in clinical trial drugs; (2) history of BTX-A, chemodenervation using phenol or alcohol, or tendon lengthening within 6 months prior to administration of the clinical trial drug or scheduled to receive any of the three during the clinical trial periods; (3) ongoing intrathecal baclofen treatment or scheduled to be treated with intrathecal baclofen during the clinical trial; (4) history of aspiration pneumonia; (5) history of causing decreased lung function; (6) patients who have any of the following: neuromuscular junction disorders, skin diseases at the site where the clinical trial drug is to be administered, dysphagia, or comorbidities unsuitable for participation in the clinical trial (severe medical illness such as malignancies, severe heart disease, and severe respiratory disease); (7) history of taking muscle relaxants, anticholinergics, benzodiazepines, aminoglycosides, lincomycins, or benzamides within 4 weeks before administration of the clinical trial drug (However, it is possible to participate if the drug that can affect muscle tone is stably taken from 4 weeks before administration of the clinical trial drug, and the dosage and usage of the drug is not expected to change during the clinical trial period.); (8) history of BTX-A within 24 weeks prior to administration of the clinical trial drug or scheduled to receive BTX-A other than the indications for this clinical trial (upper extremity muscle stiffness) during the clinical trial period; (9) tendency to bleed or on current medication with anticoagulant drugs; (10) ongoing rehabilitation treatment (physical therapy and occupational therapy) or splinting in the area where medication for clinical trials is scheduled to be administered, and a change in treatment within 4 weeks from the time of screening or expected to change the treatment during clinical trial periods; (11) severe muscle atrophy or fixed joint/muscle contracture in the area where the clinical trial drug is scheduled to be administered; (12) pregnant, breastfeeding, or a woman who tested positive for a pregnancy test at screening or plans to become pregnant during the clinical trial periods; and (13) any condition that, in the view of the investigator, would interfere with study participation.

### 6.3. Intervention

HU-014 was diluted by adding 4.0 mL of sterile normal saline (0.9% sodium chloride solution) to a vial of HU-014 (100U). Target muscles and injection doses were determined by the physician based on the degree of spasticity (Table 7) [10,11,16,17]. HU-014 was administered once to target muscles: wrist flexors, flexor carpi radialis and flexor carpi ulnaris, when the MAS score was 2 or higher, and elbow flexor (biceps brachii) and finger flexors including flexor digitorum profundus and flexor digitorum sublimis when the MAS score was 1 or higher. The injections were performed using electromyography by an experienced physician. The total dose was up to 360 U.

### 6.4. Assessment

At baseline, HU-014 was intramuscularly administered once to target muscles. Follow-up was performed at 4, 8, and 12 weeks after the injection.

### 6.5. Safety Measures (Primary Outcomes)

Adverse events were measured at every visit as non-treatment-emergent adverse events (non-TEAEs), defined as adverse events that occurred prior to administration of the HU-014, treatment-emergent adverse event (TEAE), defined as adverse events that occurred after HU-014 administration, adverse drug reaction (ADR), and serious adverse event (SAE). Patients were interviewed and asked about their symptoms or unexpected events via the telephone call in week 1 and at hospital visits in weeks 4, 8, and 12. In addition, during the study period, patients were asked to report any symptoms of discomfort or any other health problems. The severity of adverse events was classified as mild, moderate, severe, life-threatening consequences, and death. At each visit, vital signs were measured and a physical examination was performed. Laboratory tests and antibody formation tests were performed at baseline, week 4, and week 12.

### 6.6. Efficacy Measures (Secondary Outcomes)

Changes in spasticity were assessed using MAS for wrist flexor, elbow flexor, and finger flexor from baseline at 4, 8, and 12 weeks after intervention. Changes in DAS score were evaluated from baseline to 4, 8, and 12 weeks after intervention. Investigator’s Global Assessment (IGA) and Subject’s Global Assessment (SGA) were evaluated by the investigator and the patients at 4, 8, and 12 weeks after intervention. Changes in Caregiver Burden Scale (CBS) score were from baseline to 4, 8, and 12 weeks after intervention. Columbia Suicide Severity Rating Scale (C-SSRS) score covered the baseline and 4, 8, and 12 weeks after intervention.

MAS is commonly used to assess muscle spasticity, with scores ranging 0–4 [28]; here, a MAS score of 1+ was converted to 1.5. DAS is used to measure functional disability and has good reliability in patients with upper limb spasticity after stroke [29]. At baseline, the patients determined one of four domains of functional disabilities (hygiene, dressing, upper limb position, and pain) as a treatment target requiring improvement. The investigator assessed each selected item of functional disability using a four-point scale, from 0 (no disability) to 3 (severe disability); a higher score indicates a worse function [12,29]. Global assessment was used to evaluate global satisfaction on a seven-point scale (1, very dissatisfied; 2, dissatisfied; 3, somewhat dissatisfied; 4, indifferent; 5, somewhat satisfied; 6, satisfied; and 7, very satisfied). The CBS is a five-point Likert scale ranging from 0 (no difficulty) to 4 (impossible) for a total of four items: cleaning the palms of the hands, cutting fingernails, dressing, and cleaning under the armpits [30]. The C-SSRS is a questionnaire to assess risk of suicide and measures four constructs: the severity of suicidal ideation, the intensity of ideation, behavior, and lethality [31].

### 6.7. Statistical Analysis

A formal sample size calculation was not performed because this study was a phase I pilot study. The data obtained from this study will be useful for appropriate power analysis in future studies.

Statistical analysis was performed using SAS version 9.4 (SAS Institute, Cary, NC, USA). In the efficacy measure analysis, paired t-test or Wilcoxon signed-rank test was used to compare the changes in MAS, DAS, and CBS scores between the baseline and at 4, 8, and 12 weeks after the intervention. McNemar’s test was used to compare changes in C-SSRS between baseline and at 4, 8, and 12 weeks after the intervention. Statistical significance was *p* values < 0.05.

Analysis groups were classified into full analysis set (FAS), per protocol set (PPS), and safety set. FAS was intended for patients who were enrolled in this study, received the intervention, and had at least one efficacy evaluation result. PPS was applied to patients who completed the study without significant protocol violations among the FAS. Safety set was for all patients in this clinical trial who received at least one trial drug. For safety analysis, the safety set was applied, and for the efficacy analysis, FAS and PPS; the main analysis group was selected based on the FAS.

## Figures and Tables

**Figure 1 toxins-14-00730-f001:**
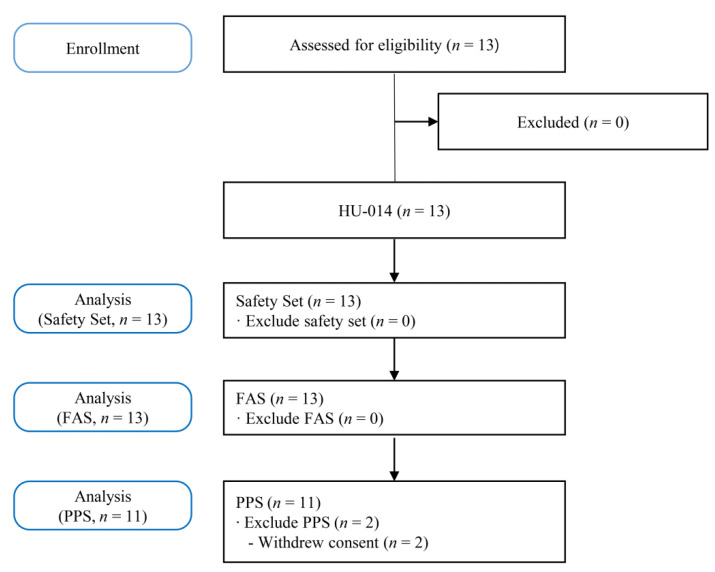
Flowchart of the study. FAS, full analysis set; PPS, per protocol set.

**Figure 2 toxins-14-00730-f002:**
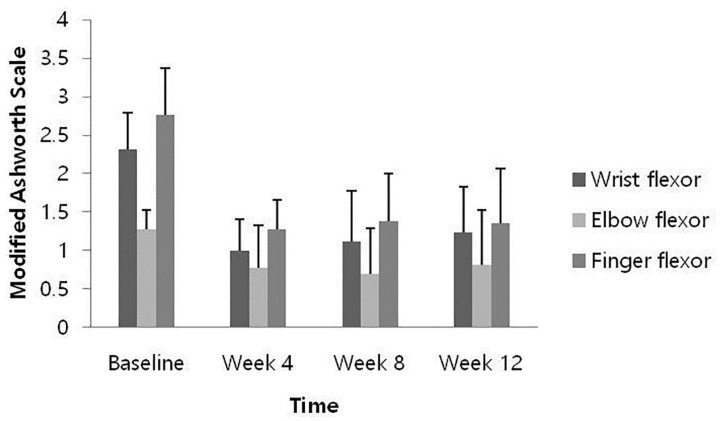
Change from baseline of muscle spasticity measured by Modified Ashworth Scale (full analysis set).

**Figure 3 toxins-14-00730-f003:**
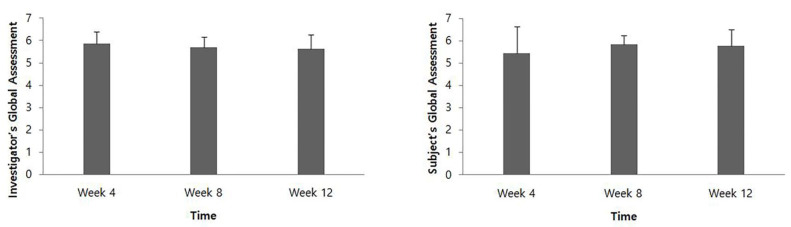
Global assessment evaluated by the investigator and the patients.

**Table 1 toxins-14-00730-t001:** Demographics and baseline characteristics of patients (full analysis set).

	HU-014 (*n* = 13)
Age (years)	50.85 ± 18.04
Sex, *n* (%)	
Male	10 (76.92)
Female	3 (23.08)
Hight (cm)	169.16 ± 8.55
Weight (kg)	69.81 ± 15.74
BMI (kg/m^2^)	24.17 ± 3.43
Concomitant therapies, *n* (%)	
Yes	10 (76.92)
No	3 (23.08)

**Table 2 toxins-14-00730-t002:** Injection dose of HU-014.

Target Muscle		Injection Dose	
Elbow flexor	Biceps brachii	n	13
		Mean ± SD	119.23 ± 32.52
		Median [Min, Max]3	100 [50.00, 150.00]
Finger flexors	Flexor digitorum profundus	n	13
		Mean ± SD	50.00 ± 0.00
		Median [Min, Max]	50 [50.00, 50.00]
	Flexor digitorum sublimis	n	13
		Mean ± SD	50.00 ± 0.00
		Median [Min, Max]	50 [50.00, 50.00]
Wrist flexors	Flexor carpi radialis	n	13
		Mean ± SD	53.08 ± 4.80
		Median [Min, Max]	50 [50.00, 60.00]
	Flexor carpi ulnaris	n	13
		Mean ± SD	48.08 ± 6.93
		Median [Min, Max]	50 [25.00, 50.00]
Total		n	13
		Mean ± SD	320.38 ± 36.66
		Median [Min, Max]	310 [250.00, 360.00]

SD, standard deviation; Min, minimum; Max: maximum.

**Table 3 toxins-14-00730-t003:** Adverse events.

System Organ Class Preferred Term	HU-014 (*n* = 13)
Ear and labyrinth disorders	
Vertigo positional ^a^	1(7.69) [2]
Metabolism and nutrition disorders	
Diabetes mellitus ^b^	1(7.69) [1]
Hyperglycemia ^b^	1(7.69) [1]
Blood and lymphatic system disorders	
Iron deficiency anemia ^c^	1(7.69) [1]
Gastrointestinal disorders	
Periodontal disease ^c^	1(7.69) [1]
Psychiatric disorders	
Insomnia ^a^	1(7.69) [1]

Data are number of patients (%) (number of cases); ^a,b,c^ The same patient presented a, b, and c. Adverse events occurred in 3 (23.08%, 7 cases) out of 13 patients.

**Table 4 toxins-14-00730-t004:** Changes from baseline of target item’s score in Disability Assessment Scale (full analysis set).

	Disability Assessment Scale (*n* = 13)	
	Mean ± SD	Median [Min, Max]
Baseline	2.08 ± 0.28	2.00 [2.00, 3.00]
Week 4	0.85 ± 0.55	1.00 [0.00, 2.00]
Change	−1.23 ± 0.60	−1.00 [−2.00, 0.00]
*p*-value *	0.0005	
Week 8	0.92 ± 0.64	1.00 [0.00, 2.00]
Change	−1.15 ± 0.69	−1.00 [−2.00, 0.00]
*p*-value *	0.0010	
Week 12	0.92 ± 0.64	1.00 [0.00, 2.00]
Change	−1.15 ± 0.69	−1.00 [−2.00, 0.00]
*p*-value *	0.0010	

SD, standard deviation; Min, minimum; Max: maximum; * Wilcoxon signed rank test.

**Table 5 toxins-14-00730-t005:** Changes from baseline in Caregiver Burden Scale (full analysis set).

Caregiver Burden Scale		Mean ± SD	Median [Min, Max]
Cleaning the palm (*n* = 13)	Baseline	1.46 ± 1.13	1.00 [0.00, 3.00]
	Week 4	1.00 ± 1.15	1.00 [0.00, 3.00]
	Change	−0.46 ± 1.33	−1.00 [−3.00, 3.00]
	*p*-value *	0.1875	
	Week 8	0.92 ± 1.04	1.00 [0.00, 3.00]
	Change	−0.54 ± 1.39	−1.00 [−3.00, 3.00]
	*p*-value *	0.1484	
	Week 12	0.92 ± 0.86	1.00 [0.00, 2.00]
	Change	−0.54 ± 1.20	0.00 [−3.00, 2.00]
	*p*-value ^†^	0.1312	
Cutting fingernails (*n* = 13)	Baseline	2.31 ± 1.25	2.00 [1.00, 4.00]
	Week 4	1.69 ± 1.49	1.00 [0.00, 4.00]
	Change	−0.62 ± 1.12	−0.00 [−3.00, 1.00]
	*p*-value *	0.1250	
	Week 8	1.38 ± 1.39	1.00 [0.00, 4.00]
	Change	−0.92 ± 1.12	−1.00 [−3.00, 1.00]
	*p*-value ^†^	0.0114	
	Week 12	1.38 ± 1.04	1.00 [0.00, 4.00]
	Change	−0.92 ± 1.19	0.00 [−3.00, 0.00]
	*p*-value *	0.0313	
Dressing (*n* = 13)	Baseline	1.23 ± 0.83	1.00 [0.00, 2.00]
	Week 4	1.00 ± 1.00	1.00 [0.00, 3.00]
	Change	−0.23 ± 0.93	0.00 [−2.00, 2.00]
	*p*-value *	0.5625	
	Week 8	0.85 ± 0.69	1.00 [0.00, 2.00]
	Change	−0.38 ± 0.51	0.00 [−1.00, 0.00]
	*p*-value *	0.0625	
	Week 12	0.85 ± 0.80	1.00 [0.00, 2.00]
	Change	−0.38 ± 0.77	0.00 [−2.00, 1.00]
	*p*-value *	0.1875	
Cleaning under the armpit (*n* = 13)	Baseline	1.46 ± 0.66	2.00 [0.00, 2.00]
	Week 4	0.92 ± 0.76	1.00 [0.00, 2.00]
	Change	−0.54 ± 0.78	0.00 [−2.00, 0.00]
	*p*-value *	0.0625	
	Week 8	1.15 ± 0.90	1.00 [0.00, 3.00]
	Change	−0.31 ± 0.95	0.00 [−2.00, 2.00]
	*p*-value *	0.4063	
	Week 12	1.00 ± 0.82	1.00 [0.00, 2.00]
	Change	−0.46 ± 0.97	0.00 [−2.00, 1.00]
	*p*-value ^†^	0.1111	

SD, standard deviation; Min, minimum; Max: maximum; * Wilcoxon signed-rank test; ^†^ paired *t*-test.

**Table 6 toxins-14-00730-t006:** Changes from baseline suicidal ideation in Columbia Suicide Severity Rating Scale (full analysis set).

Columbia Suicide Severity Rating Scale	HU-014 (*n* = 13)		
	Baseline → Week 4, *n* (%)	Baseline → Week 8, *n* (%)	Baseline → Week 12, *n* (%)
Low → low	13 (100)	13 (100)	13 (100)
Low → Moderate/high	0	0	0
Moderate/high → low	0	0	0
Moderate/high → Moderate/high	0	0	0
*p*-value *	Not calculated	Not calculated	Not calculated

* Testing for change within-treatment group (McNemar’s test).

**Table 7 toxins-14-00730-t007:** Recommended injection doses and sites for upper limb muscles.

Target Muscle	Recommended Injection Dose (U)	Number of Injection Sites
Biceps brachii	100–200	Up to 4
Flexor carpi radialis	15–60	1–2
Flexor carpi ulnaris	10–50	1–2
Flexor digitorum profundus	15–50	1–2
Flexor digitorum sublimis	15–50	1–2

## Data Availability

The data used and/or analyzed in this study are available from the corresponding author upon reasonable request.

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
