# Peer review of "Safety and Efficacy of HU-014 in the Treatment of Post-Stroke Upper Limb Spasticity: A Phase I Pilot Study"

_toxins, 2022, doi:10.3390/toxins14110730_

Round 1
Reviewer 1 Report
The article presents the results of phase 1 clinical trials of a new drug botulinum toxin type A. The introduction is written completely enough, the methods used are sufficient to describe the results obtained. Adequate statistical methods were used. The results are sufficient to confirm the safety of the new BoNT/A formulation. The discussion confirms the results obtained.
The main problem is whether there is enough evidence to argue for the effectiveness of the compound. The "Limitations of the Study" section does not answer this question. The patient group used is too small. Sample size calculations are not given. Paragraph 2.5 mentions that samples have been taken for antibody formation studies, but the results of these studies are not available.
Table 1 shows the recommended injection doses. But it is necessary to specify - Doses Used - otherwise there is a question of what doses were actually used and why these particular doses are recommended. The variation in the doses used in the phase 1 trials can be used in a dose range study. However, there is no such data in the article.
The article needs to be revised to properly emphasize the safety study of the use of the new drug.
Author Response
Dear Editors and Reviewers:
Thank you very much for considering our manuscript (Manuscript ID toxins-1926945) entitled “Safety and efficacy of HU-014 in the treatment of post-stroke upper limb spasticity: A phase I clinical trial” provisionally for publication. We are grateful for your insightful comments and suggestions.
This letter outlines our point-by-point responses to the editor’s and reviewers’ suggestions, requests, and questions. We have thoroughly read all of the comments provided by the reviewers and have attempted to follow their advice as much as possible. We hope our efforts meet your expectations.
The manuscript has also been carefully reviewed by an experienced editor whose first language is English and who specializes in editing papers written by scientists whose native language is not English.
Thank you once again for accepting our manuscript provisionally for publication in Toxins. We look forward to receiving your decision.
Yours sincerely,
Reviewers’ Comments and Responses
Reviewer #1:
The article presents the results of phase 1 clinical trials of a new drug botulinum toxin type A. The introduction is written completely enough, the methods used are sufficient to describe the results obtained. Adequate statistical methods were used. The results are sufficient to confirm the safety of the new BoNT/A formulation. The discussion confirms the results obtained.
The main problem is whether there is enough evidence to argue for the effectiveness of the compound. The "Limitations of the Study" section does not answer this question. The patient group used is too small. Sample size calculations are not given. Paragraph 2.5 mentions that samples have been taken for antibody formation studies, but the results of these studies are not available.
Response: Thank you for raising this important point. We completely agree that the sample size of this study is too small, which may have reduced the statistical power of measuring the outcomes. We have added an explanation that this study was a pilot study to the Study design section of the Materials and Methods and Study limitation section of the Discussion (Lines 66 and 323–324). Please understand that as this study was a phase 1 pilot study, no formal sample size calculation was performed and the sample size was kept minimal. A future study with a large sample size is needed to sufficiently demonstrate the effectiveness of HU-104. Therefore, we plan to conduct phase II or III studies on the safety and efficacy of HU-014 with a larger sample size and long-term follow-up. We have stated this in the statistical analysis section of the Materials and Methods and the Study limitation section of the Discussion (Lines 153–154, 323–325, and 330–331).
Regarding the results of antibody formation tests, as mentioned in the Safety Measures section of Results, the results of the antibody formation test confirmed that no patients were “positive” at any time point (Lines 210–211).
Table 1 shows the recommended injection doses. But it is necessary to specify - Doses Used - otherwise there is a question of what doses were actually used and why these particular doses are recommended. The variation in the doses used in the phase 1 trials can be used in a dose range study. However, there is no such data in the article.
Response: Thank you for your valuable comment. The recommended injection doses and sites used in this study followed those of previous studies on the efficacy and safety of botulinum toxin type A in the treatment of post-stroke upper limb spasticity [1-4]. Consistent with the methods employed in previous studies, the dose for injection and the target muscles were selected by the physician based on the degree of spasticity. The actual dose of HU-104 injected in this study is shown in Table 3. The injection dose varies depending on the patient’s spasticity. However, this study is not a dose-range study (i.e., it is not a clinical trial in which different doses of an agent are tested to determine which dose is most effective and/or least harmful). To clarify this issue, we have revised the text in the Intervention section of Materials and Methods and added references of previous studies. We have also added this point in the Discussion (Lines 106, 109–110, and 276–280).
The article needs to be revised to properly emphasize the safety study of the use of the new drug.
Response: Thank you for raising this important point. To emphasize the safety study of HU-014, descriptions have been added to the Safety Measures section in the Results and the Discussion (Lines 118, 125–126, 186–190, and 268–273).
Reviewer 2 Report
Manuscript: toxins-1926945
Title: Safety and efficacy of HU-014 in the treatment of post-stroke upper limb spasticity: A phase I clinical trial
This article contains interesting data from a small size cohort in a phase I clinical trail design. Thirteen stroke survivors in the chronic phase with upper limb spasticity signs were injected with a new BoNT-A product HU-014, licenced for cosmetic use in Korean. Complete data set on safety is available form 13 stroke survivors.
The primary outcome of this phase I clinical trial was safety, and this showed no notable safety-related issues to the HU-o14 application in reported adverse events, vital signs, physical examination, laboratory test, and antibody formation test. This study therefore is a positive study.
The secondary outcomes from baseline onward (prospectively, measuring at weeks 4, 8, and 12) in a non-placebo-controlled fashion monitored impairment (changes in MAS score for wrist, elbow, and finger flexor), subjective response to treatment, activity and disability level in the upper limb (DAS; IGA and SGA; CBS) and mood/affective situation while the study periode (C-SSRS).
Major points of criticism:
Data is interesting but contains only complete data from a small cohort of 11 patients in the chronic stage of spasticity following stroke and therefore represents first data on safety in limb injections of HU-014 in chronic stroke survivors. Authors may claim to publish first safety data in post stroke patients in a phase I design in higher dose per session than in cosmetic use.
With respect to other studies in the field of primary endpoint safety (phase III study with cohort of more than 100 patients, e.g. the TOWER study, Wissel et al., Neurology), this evaluation follows the concept of a phase I study and only gain reported side effects and not search for side effects: e.g. 1. interviewing the patients and asks whether symptoms appear as clinical signs that could represent side effects (e.g. swallowing problems or breathing problems) and 2. apply specific testing for clinical symptoms of side effects (e.g. testing breathing capacity and searching for signs of toxin spread to adjacent muscles by testing muscle strength). The different concepts should be discussed.
As it is a small cohort and not a placebo-controlled design we do not have enough reliable information on efficacy and therefore data represents a positive signal in a small cohort study. This imply this are first data and authors cannot claim to show reliable efficacy data of HU-014.
Having this in mind the title of the paper is misleading as title suggests the paper confirm a good safty profil in a large cohort and reliable efficacy data of HU-014. With respect to efficacy data paper shows significant changes in standard parameters measured in a prospective cohort study in upper limb spasticity treatment of chronic stroke survivors.
Minor point of criticism
line 23/24: authors site 20 years and 10 years old papers, citation of more recent publications, e.g. systematic reviews published in TOXINS on "spasticity incidence or prevalence" following stroke is more adequate and recommended.
Authors should make a statement what definition of spasticity they are following in their design. Authors should comment on the important differentiation between increased muscle tone (velocity dependend increase in muscle tone = MAS) and disabling spasticity (e.g. spasticity that does need treatment, MAS >= 2).
line 90/91 Dilution of 100 units HU-014 was done with 4 ml isothon NaCl, therefore it is a concentration of 25 units / ml, which is different than dilutions used in pivotal studies with ona- or inco-BoNT-A. Authors should discuss the implications using a higher dilution of BoNT-A in this phase I trial.
Intrestingly HU-014 dosing is using the same "unit-base in vials" than ona- and inco-BoNT-A. Authors should explain whether this definition of units is based on animal data or human experiments.
Summery
This present article is interesting as it captures first safety data of a treatment of higher dosing than in cosmetic use and injection in limb muscles in the model of chronic upper limb spasticity with a new product containing BoNT-A, up to now only registered for cosmetic use in Korea. Major critisism: As it is not a placebo-controlled double-blind study in a small cohort (n= 11) authors can not classify it as a study showing reliable "efficacy data" of the product.
Author Response
Dear Editors and Reviewers:
Thank you very much for considering our manuscript (Manuscript ID toxins-1926945) entitled “Safety and efficacy of HU-014 in the treatment of post-stroke upper limb spasticity: A phase I clinical trial” provisionally for publication. We are grateful for your insightful comments and suggestions.
This letter outlines our point-by-point responses to the editor’s and reviewers’ suggestions, requests, and questions. We have thoroughly read all of the comments provided by the reviewers and have attempted to follow their advice as much as possible. We hope our efforts meet your expectations.
The manuscript has also been carefully reviewed by an experienced editor whose first language is English and who specializes in editing papers written by scientists whose native language is not English.
Thank you once again for accepting our manuscript provisionally for publication in Toxins. We look forward to receiving your decision.
Yours sincerely,
Reviewers’ Comments and Responses
Reviewer #2:
Manuscript: toxins-1926945
Title: Safety and efficacy of HU-014 in the treatment of post-stroke upper limb spasticity: A phase I clinical trial
This article contains interesting data from a small size cohort in a phase I clinical trail design. Thirteen stroke survivors in the chronic phase with upper limb spasticity signs were injected with a new BoNT-A product HU-014, licenced for cosmetic use in Korean. Complete data set on safety is available form 13 stroke survivors.
The primary outcome of this phase I clinical trial was safety, and this showed no notable safety-related issues to the HU-o14 application in reported adverse events, vital signs, physical examination, laboratory test, and antibody formation test. This study therefore is a positive study.
The secondary outcomes from baseline onward (prospectively, measuring at weeks 4, 8, and 12) in a non-placebo-controlled fashion monitored impairment (changes in MAS score for wrist, elbow, and finger flexor), subjective response to treatment, activity and disability level in the upper limb (DAS; IGA and SGA; CBS) and mood/affective situation while the study periode (C-SSRS).
Major points of criticism:
Data is interesting but contains only complete data from a small cohort of 11 patients in the chronic stage of spasticity following stroke and therefore represents first data on safety in limb injections of HU-014 in chronic stroke survivors. Authors may claim to publish first safety data in post stroke patients in a phase I design in higher dose per session than in cosmetic use.
With respect to other studies in the field of primary endpoint safety (phase III study with cohort of more than 100 patients, e.g. the TOWER study, Wissel et al., Neurology), this evaluation follows the concept of a phase I study and only gain reported side effects and not search for side effects: e.g. 1. interviewing the patients and asks whether symptoms appear as clinical signs that could represent side effects (e.g. swallowing problems or breathing problems) and 2. apply specific testing for clinical symptoms of side effects (e.g. testing breathing capacity and searching for signs of toxin spread to adjacent muscles by testing muscle strength). The different concepts should be discussed.
Response: Thank you for your comment. The patients were interviewed and asked about their symptoms or unexpected events over a telephone call in week 1 and at hospital visits in weeks 4, 8, and 12. During the study period, the patients were asked to report any symptoms of discomfort or any other health problems. When the adverse events required specific testing, we performed it. However, in this study, no adverse effects such as swallowing problems or breathing problems were observed, except the adverse reactions shown in Table 4, so no specific test such as that of breathing capacity was conducted. Nevertheless, in the TOWER study conducted by Wissel et al., safety assessments included measurements of pulmonary functions such as forced expiratory volume in 1 second and maximal inspiratory pressure, but these tests were not included as safety measures in this study. This point has been added to the Safety measures section of the Materials and methods and the Study limitation section of the Discussion (Lines 122–125 and 331–335).
As it is a small cohort and not a placebo-controlled design we do not have enough reliable information on efficacy and therefore data represents a positive signal in a small cohort study. This imply this are first data and authors cannot claim to show reliable efficacy data of HU-014.
Response: Thank you for raising this crucial point. As mentioned in the response to reviewer #1 above, we completely agree that the sample size of this study is too small, which may have reduced the statistical power of measuring the outcomes. We have added an explanation that this study was a pilot study to the Study design section of the Materials and Methods and Study limitation section of the Discussion (Lines 68 and 323–324). Please understand that as this study was a phase 1 pilot study, the sample size was kept minimal. A future study with a large sample size is needed to sufficiently demonstrate the effectiveness of HU-104. Therefore, we plan to conduct phase II or III studies on the safety and efficacy of HU-014 with a larger sample size and long-term follow-up. We have stated this in the statistical analysis section of the Materials and Methods and the Study limitation section of the Discussion (Lines 153–154, 323–325, and 330–331).
Having this in mind the title of the paper is misleading as title suggests the paper confirm a good safty profil in a large cohort and reliable efficacy data of HU-014. With respect to efficacy data paper shows significant changes in standard parameters measured in a prospective cohort study in upper limb spasticity treatment of chronic stroke survivors.
Response: We completely agree with your opinion. This study was a phase I pilot study with a small sample size. To avoid the confusion that you mentioned, we have slightly modified the title of the paper to “Safety and efficacy of HU-014 in the treatment of post-stroke upper limb spasticity: A phase I pilot study” (Lines 1–2).
Minor point of criticism
line 23/24: authors site 20 years and 10 years old papers, citation of more recent publications, e.g. systematic reviews published in TOXINS on "spasticity incidence or prevalence" following stroke is more adequate and recommended.
Response: Thank you for your comment. We have updated the old references to more recently published articles regarding the incidence or prevalence of spasticity after stroke [5,6] (Lines 37–39).
Authors should make a statement what definition of spasticity they are following in their design. Authors should comment on the important differentiation between increased muscle tone (velocity dependend increase in muscle tone = MAS) and disabling spasticity (e.g. spasticity that does need treatment, MAS >= 2).
Response: Thank you for your comment. Spasticity is a motor disorder characterized by a velocity-dependent increase in muscle tone or a tonic stretch reflex associated with hypertonia [7]. The evaluation of spasticity in this study includes assessment of the velocity-dependent increase of resistance to passive movements. We measured spasticity using the modified Ashworth Scale (MAS), which is the most commonly used tool for quantifying spasticity [8]. Regarding disabling spasticity, there is no generally accepted definition of this condition [6]. Disabling spasticity was defined by Lundström as spasticity that requires interventions such as physiotherapy and orthoses or pharmacological treatment. It was also defined as spasticity that affects movement function, activity performance, or participation in social life, accompanied by positive symptoms of upper motor neuron syndrome [6]. Therefore, disabling spasticity requires a comprehensive clinical evaluation of patients with regard to the presence and impact of spasticity from a disability perspective. The MAS used to measure spasticity in this study has the limitation that it measures only increased muscle tone. Disability spasticity was not directly evaluated in this study, but we assessed functional disabilities by including DAS as an outcome measure. A description of this has been added to the Discussion (Lines 308–320).
line 90/91 Dilution of 100 units HU-014 was done with 4 ml isothon NaCl, therefore it is a concentration of 25 units / ml, which is different than dilutions used in pivotal studies with ona- or inco-BoNT-A. Authors should discuss the implications using a higher dilution of BoNT-A in this phase I trial.
Response: Thank you for raising this key point. A wide range of dilution ratios may be used for post-stroke upper limb spasticity [9]. The dilution of 25 units/mL of HU-014 was used in this study. For onabotulinum toxin A, the dilution of 50 units/mL was most commonly used, but higher dilutions of 25 units/mL and 20 units/mL were also used in several studies [10,11]. Various dilutions such as 20, 50, and 100 units/mL were also used for incobotulinum toxin A [12]. It is assumed that a higher dilution of botulinum toxin A may result in better therapeutic effects than a lower dilution volume; however, this point remains controversial [10-12]. Therefore, the implications for using higher dilutions of 25 units/mL of HU-014 than those in pivotal studies with onabotulinum toxin A or incobotulinum toxin A were unclear. Further research is needed to identify the implications of using a higher dilution of botulinum toxin type A. We have added a description of this in the Discussion (Lines 283–291).
Intrestingly HU-014 dosing is using the same "unit-base in vials" than ona- and inco-BoNT-A. Authors should explain whether this definition of units is based on animal data or human experiments.
Response: Botulinum toxin dose is measured in units (U), based on the median lethal dose (LD50) of the neurotoxin. 1 unit of botulinum toxin is the dose of intraperitoneally injected toxin found to kill 50% (LD50) of a group of mice. The biological potency of HU-014 was based on the LD50 after the mouse LD50 assay. (Lines 52–55).
Summary
This present article is interesting as it captures first safety data of a treatment of higher dosing than in cosmetic use and injection in limb muscles in the model of chronic upper limb spasticity with a new product containing BoNT-A, up to now only registered for cosmetic use in Korea. Major critisism: As it is not a placebo-controlled double-blind study in a small cohort (n= 11) authors can not classify it as a study showing reliable "efficacy data" of the product.
Response: Thank you for raising this important point. As mentioned in the response to the reviewer above, we completely agree that the sample size of this study is small, which may have reduced the statistical power of measuring the outcomes. We have added an explanation that this study was a pilot study to the title, Study design section of the Materials and Methods, and Study limitation section of the Discussion (Lines 1–2, 68, and 323–324). Please understand that as this study was a phase 1 pilot study, the sample size was kept minimal. A future study with a large sample size is needed to sufficiently demonstrate the effectiveness of HU-104. Therefore, we plan to conduct phase II or III studies on the safety and efficacy of HU-014 with a larger sample size and long-term follow-up. We have stated this in the Study limitation section of the Discussion (Lines 153–154, 323–325, and 330–331).
Round 2
Reviewer 2 Report
Thank you for revisions and covering letter, paper is improved and with minor changes qualified for publication in Toxins.
Major change recommended: Citation of Wissel et al. TOWER study is missing in the References.
Recommended changes in content before publishing in Toxins:
312-313: Disability from spasticity was not directly evaluated in this study, but changes in passive function were assessed by including DAS as an secondary outcome measure.
326-330: Finally, in this study only reported adverse effects were considered and this study did not search for additional adverse effects with a check list, like the TOWER study, a dose escalation study with inco-BoNT did. . As well this study did not include measurements of pulmonary function, like the TOWER study, such as forced expiratory volume in 1 second and maximal inspiratory pressure (Wissel et al., TOWER study)
Minor changes necessary
274: units/mLand 20 units/mLwere
275: units/mLwere
278: units/mLof
Author Response
Dear Editorand Reviewers:
Thank you very much for considering our manuscript (Manuscript ID toxins-1926945) entitled “Safety and efficacy of HU-014 in the treatment of post-stroke upper limb spasticity: A phase I clinical trial” provisionally for publication. We are grateful for your insightful comments and suggestions.
This letter outlines our point-by-point responses to the editor’s and reviewers’ suggestions, requests, and questions. We have thoroughly read all of the comments provided by the reviewers and have attempted to follow their advice as much as possible. We hope our efforts meet your expectations.
The manuscript has also been carefully reviewed by an experienced editor whose first language is English and who specializes in editing papers written by scientists whose native language is not English.
Thank you once again for accepting our manuscript provisionally for publication in Toxins. We look forward to receiving your decision.
Yours sincerely,
Reviewers’ Comments and Responses
Reviewer #2:
Thank you for revisions and covering letter, paper is improved and with minor changes qualified for publication in Toxins.
Major change recommended: Citation of Wissel et al. TOWER study is missing in the References.
Response: Thank you for raising this important point. The Wissel et al. TOWER study has been cited in the References (Lines 336 and 463–465).
Recommended changes in content before publishing in Toxins:
312-313: Disability from spasticity was not directly evaluated in this study, but changes in passive function were assessed by including DAS as an secondary outcome measure.
Response: Thank you for your comment. The sentence has been changed as you recommended (Lines 319–321).
326-330: Finally, in this study only reported adverse effects were considered and this study did not search for additional adverse effects with a check list, like the TOWER study, a dose escalation study with inco-BoNT did. . As well this study did not include measurements of pulmonary function, like the TOWER study, such as forced expiratory volume in 1 second and maximal inspiratory pressure (Wissel et al., TOWER study)
Response: Thank you for your comment. The sentences has been changed as you recommended (Lines 332–336).
Minor changes necessary
274: units/mLand 20 units/mLwere
Response: Thank you for your comment. Spaces were added between the words (Line 285).
275: units/mLwere
Response: Thank you for your comment. A space was added between the words (Line 286).
278: units/mLof
Response: Thank you for your comment. A space was added between the words (Line 289).